# Work Environment and Socio-Demographic Factors of Psychiatric Nurses: A Cross-Sectional Study in Hospitals of Eastern Saudi Arabia

**DOI:** 10.3390/jcm13216506

**Published:** 2024-10-30

**Authors:** Husain A. Al Shayeb, Ahmad E. Aboshaiqah, Naif H. Alanazi

**Affiliations:** 1College of Nursing, King Saud University, Riyadh 11451, Saudi Arabia; hualshayeb@moh.gov.sa (H.A.A.S.); aaboshaiqah@ksu.edu.sa (A.E.A.); 2Nursing Department, Erada and Mental Health Complex, Dammam 32312, Saudi Arabia

**Keywords:** work environment, nurses, psychiatric hospitals, mental health, Saudi Arabia

## Abstract

The work environment in healthcare institutions, especially in psychiatric hospitals, plays a crucial role in shaping the experiences and efficacy of nurses’ performance. This environment is influenced by various factors such as facility design, resource availability, workplace culture, support systems, and interpersonal dynamics. Understanding the intricate dynamics of the work environment in psychiatric hospitals is essential for ensuring the provision of high-quality mental healthcare services and enhancing the overall quality of life for both patients and healthcare providers, including nurses. However, the work environment of psychiatric nurses in the eastern region of Saudi Arabia is still inadequately understood. **Objective:** This study aimed to examine the work environment of nurses working in psychiatric hospitals in the eastern region of Saudi Arabia. **Method:** A cross-sectional research design was employed on a sample of 346 nurses using a non-probability convenience sampling technique. The survey method was adopted with the Practice Environment Scale of the Nursing Work Index (revised, Arabic version). **Results:** The study found a significant association between psychiatric nurses’ work environment and their socio-demographic characteristics. Demographic factors, such as age and years of experience, were identified as influencing factors of nurses’ perceptions of their work environment. Younger nurses and those with fewer years of experience reported greater job satisfaction when their work environment was positive. **Conclusions:** This study underscores the critical importance of maintaining a supportive work environment for psychiatric nurses due to its possible direct influence on their job satisfaction, work performance, quality of life, and overall well-being. Tailoring interventions to address demographic variations in perceptions of the work environment can enhance the well-being of nurses and improve the quality of care provided to psychiatric patients. These findings contribute to the body of knowledge on psychiatric nursing and have clinical implications for healthcare institutions that aim to optimize their work environments and retain a skilled and satisfied nursing workforce.

## 1. Introduction

The work environment in healthcare settings, including psychiatric hospitals, plays an instrumental role in shaping the experiences and performance of nursing professionals [1]. It encompasses a spectrum of factors that range from physical aspects, such as facility design and resource availability, to intangible elements, such as workplace culture, support systems, and interpersonal dynamics [2]. Understanding the nuanced dynamics of the work environment in psychiatric hospitals is essential because it has profound implications for the delivery of mental healthcare and overall quality of life for healthcare providers and patients [3].

Clinical practices in different disciplines are based on diverse healthcare services. Nurses frequently experience a significant workload, facing increased physical, social, psychological, and environmental challenges [4]. Addressing these challenges is crucial for patient safety and security, necessitating appropriate interventions, effective management, and stratified services [5]. Moreover, a heavy workload and long shift hours increase work-related conflicts, supervisory conflicts, low-level organizational support, and issues that lead to nursing staff shortages [6,7]. Long work hours, particularly in critical departments, result in increased absences due to sickness; a higher rate of staff turnover; deteriorating psychological, physical, and social health; and burnout [8]. In the case of psychiatric clinics, precautionary measurements, prevention, diagnosis, and treatment are based on various maladaptive behaviors [9], faulty schemes [10], and cognitive distortions [11] that must be addressed.

In the Kingdom of Saudi Arabia (KSA), the first mental hospital was established in 1952 by the Ministry of Health (MOH) with all the potential equipment [12]. The country has 21 highly equipped mental health hospitals with a capacity of more than 4000 patients for treating specialized mental health illnesses [13]. Healthcare professionals, particularly nurses, are appointed annually based on their high emotional stability and ability to provide comprehensive care to patients with mental health problems [14]. The MOH of the KSA postulates the act that protects the health and dignity of healthcare workers (HCWs). Therefore, certain rules and regulations have been developed for HCWs who work in critical psychiatric departments based on their effort and healthcare scenarios [15]. For example, Article 31 of the Saudi government’s basic constitution law [16,17] and a similar article about the health law [18] serve as sources of mental health constitutional laws that have been organized to strengthen healthcare services and reduce the stigmatization of nurses working with mentally ill patients [19]. On the other hand, the Saudi Commission for Health Specialties works on exclusive plans about health transformational intuition, including collaborations with drug and food authorities, healthcare regulation, and health profession practice laws to supply primary psychological care to patients [20]. In addition, the MOH of the KSA has issued guidelines and identified specific standards and challenges in nursing care for psychiatric patients, ensuring compliance with international standards [21].

Working in the psychiatric department is characterized by lethargy and hardships because it not only affects the mental health of nurses but also their physical, physiological, and personal lives [22]. According to the World Health Organization (WHO), dealing with psychiatric patients requires effective monitoring and proper environmental establishment for healthcare workers [23]. The basic perspective is associated with the working environment, including social features, physical condition, reputation, and setting [24]. In accordance with workplace environment theory, employees working in an organization depend on five cues and essential features that affect working conditions, capacity, capability, efficacy, and job performance. These factors are temperature, air, color or light, space, and sound [25]. In accordance with this theory, using environmental cues and perspectives considerably affects an entity working in a particular department. It states that nurses’ treatment of psychiatric patients requires the effective healthcare management of the nurses themselves [26]. This theory is also associated with nursing healthcare services, suggesting that environmental care is not only for patients but also for nurses to gain an effective positive and optimistic outlook based on environmental harmony [27]. This theory is immensely applicable to the recovery of patients, the protection of nurses’ well-being, and the establishment of harmony between environmental cues and intrinsic individual conditions [28].

In the KSA, the nursing board committee was organized in 2019 during the beginning of the COVID-19 pandemic [29]. The primary agenda of this committee was to reestablish and identify various healthcare challenges during the pandemic. The committee states that psychiatric nurses are still inclusively separated from the West to protect themselves and reduce work overload due to the COVID-19 pandemic’s new and highly advanced types of machinery, equipment, and interventions [30]. Psychiatric nurses usually ponder upon the health-related parameters of psychiatric patients. Negativity and hyperactivity in their behavior are typically portrayed under environmentally hazardous conditions [31].

In the field of mental healthcare, individuals dealing with psychiatric disorders frequently face challenges, including stigma, discrimination, and limited access to appropriate services. Many countries have dealt with deep-rooted societal attitudes toward mental health, which have historically fostered a culture of silence and misconceptions that surround these conditions [32]. Similar to many other nations, the KSA has faced considerable challenges from society toward patients with mental health issues. This prevailing environment has frequently led to delayed diagnoses and inadequate treatment for those in need. Nurses in psychiatric hospitals play a pivotal role in addressing these issues. They are the frontline care providers, offering not only medical treatment but also emotional support and understanding to individuals with mental health conditions [33]. However, this nursing specialization has its own unique set of challenges. Psychiatric nurses frequently encounter emotionally charged and unpredictable situations, which can be physically and mentally taxing. Moreover, they contend with deeply ingrained societal biases and misunderstandings about mental illness; these factors can affect their job satisfaction, overall well-being, and, ultimately, their effectiveness in providing quality care [34].

Previous studies have consistently highlighted the importance of understanding the influence of the work environment on nurses, particularly in the context of mental healthcare. For example, research in Western countries has shown that a supportive work environment positively correlates with the job satisfaction of nurses and their ability to deliver high-quality care to individuals with psychiatric disorders. Moreover, a conducive work atmosphere has been linked to lower levels of burnout and emotional exhaustion among psychiatric nurses [35]. However, within the specific context of the KSA, and specifically, its eastern region, research concerning the effect of the work environment on nurses in psychiatric settings is scarce. This knowledge gap is particularly pronounced in a cultural and societal context distinct from Western healthcare systems, where attitudes, practices, and healthcare infrastructure may vary significantly [32,36]. Consequently, this research void must be filled immediately, because it hinders the ability to effectively address the challenges faced by psychiatric nurses and individuals with mental health conditions. Moreover, this gap must be filled in the eastern region of the KSA because the nurses working in psychiatric hospitals in this region are facing multiple challenges in terms of stress, job satisfaction, and career path improvement.

Therefore, the current study assumes paramount significance. It aimed to bridge this research gap by comprehensively examining how the work environment influences psychiatric nurses within the unique cultural, societal, and healthcare context of the eastern region of the KSA. By doing so, it not only contributes to the global body of knowledge on mental health nursing but also provides insights that can directly inform healthcare practices, policies, and public perception in KSA. Ultimately, this research aspired to enhance the quality of mental healthcare services, ensure the well-being of psychiatric nurses, and promote a more empathetic and supportive environment for individuals dealing with mental health challenges in this region. Hence, the current study aimed to assess the work environment for nurses in psychiatric hospitals in the eastern region of the KSA. The following objectives helped to achieve the aim of this research: (1) identify the associations of their work environment in psychiatric hospitals, and (2) identify differences in the perceptions of work environment among nurses working in the two selected mental health settings. These objectives were guided by the following research hypotheses:There are associations between nurses’ socio-demographic characteristics and perceptions of their work environment in psychiatric hospitals.There are differences in the perceptions of their work environment when psychiatric nurses are grouped according to their socio-demographic characteristics.

## 2. Materials and Methods

### 2.1. Research Design

This quantitative study used a cross-sectional, correlational research design. This research is also quantitative in nature. Thus, this design is most suitable because the selection of a particular research design is driven by the careful alignment between the design and research objectives and the inherent nature of the research question. The suitability of a design is also influenced by factors, such as resource availability, ethical considerations, previous research in the field, type of data required, research context and setting, time frame, complexity of the phenomenon under investigation, and practicality of implementation. The most suitable research design emerges as the one that harmonizes with these critical considerations, ensuring that the chosen research method is not only academically rigorous but also feasible and ethically sound, ultimately enhancing the study’s capacity to effectively address the research question or hypothesis.

### 2.2. Study Setting, Population, and Sampling

This study was conducted in two psychiatric hospitals situated in different cities in the eastern region of the KSA. These hospitals were deliberately chosen because of their robust mental health department setups, which encompassed a wide range of specialized services and essential equipment. The exact number of nurses in each hospital varied, which was contingent on the hospital’s size and patient capacity. In such psychiatric hospitals, maintaining a substantial nursing staff is common to ensure round-the-clock care and support for patients. 

In Hospital A, the number of nurses is about 395 (male nurses = 215 and female nurses = 180). This hospital has three inpatient departments for male patients, three inpatient departments for female patients, an emergency department, and an outpatient department. In Hospital B, the number of nurses is about 420 (male nurses = 303 and female nurses = 117). The complex includes three inpatient departments for male addict patients, three rehabilitation departments for male addict patients, three inpatient departments for male psychiatric patients, one rehabilitation department for male psychiatric patients, two inpatient departments for psychiatric female patients, one inpatient department for female addict patients, an emergency department, and an outpatient department. The patient population served by these hospitals is diverse, comprising individuals with a wide array of mental health disorders, including, but not limited to, depression, anxiety disorders, bipolar disorder, schizophrenia, personality disorders, and substance use. The two distinct hospital settings offer a valuable opportunity to explore the associations of the work environment of nurses in psychiatric care across different geographical cities, potentially shedding light onto regional variations and best practices within the field.

This study used a non-probability-based convenience sampling technique among nurses working in psychiatric settings in the selected hospitals. Using the G-force sample calculation method, as shown below, the estimated sample size was 300 nurses, with 150 from each hospital. 

The formula for sample size is as follows:Sample Size Formula = [z^2^ × p(1 − p)]/e^2^/1 + [z^2^ × p(1 − p)]/e^2^ × N]
where nurses’ population in the two hospitals = N = 1000;

Z-score *=* z = 2.58;

Margin of error *=* e = 0.05;

Standard deviation *=* p = 0.5;

Sample size = [z^2^ × p(1 − p)]/e^2^/1 + [z^2^ × p(1 − p)]/e^2^ × N] = [2.582 × 0.5(1 − 0.5)]/0.052/1 + [2.582 × 0.5(1 − 0.5)]/0.052 × 1500] = [6.6564 × 0.25]/0.0025/1 + [6.6564 × 0.25]/1.0625] = 713/2.5663 = 278 approximately equal to 300.

Therefore, the appropriate sample size was 300.

Inclusion Criteria

Nurses who were willing to participate;Nurses who have experience working with psychiatric patients for at least 2 years because of the expertise required for managing the appropriate treatment and management of psychiatric patients;Only clinical, registered, and manager nursing staff were included;Only nurses who deal with psychiatric patients were included.

Exclusion Criteria

Nurses aged below 22 years or above 60 years were excluded from this study to ensure a more homogeneous sample that focused on nurses in mid-career stages. Younger nurses below 22 years are frequently in the early phases of their education or career, while those above 60 years may be nearing retirement or experiencing different career motivations. By narrowing the age range, this study aimed to capture insights and perspectives from nurses who are likely to be actively engaged in their careers, thus enhancing the coherence of the research findings and minimizing potential confounding factors related to career stage and life circumstances.

### 2.3. Data Collection Procedure

Before the start of the study, written permission was acquired from the regional ethical review board and the College of Nursing at King Saud University by providing a brief research proposal. Thereafter, with the Institutional Review Board (IRB) approval from the regional health directorate, data collection permission was also obtained from the MOH in accordance with its rules and policies. Written informed consent was obtained from the participants by briefly mentioning the study’s goal. Participants could join the study by signing the informed consent form.

The filling of the questionnaire fully depended on the willingness of the participants. Filling the form took only 5–10 min. The researcher collected the data, assigned coding, and then stored them in a password-protected drive. Further data analysis was performed in accordance with the research requirements.

### 2.4. Measurements/Tools 

Quantitative research typically uses a survey-based questionnaire to collect data. In this study, the researcher developed a survey-based data collection form that comprised the following tools in Arabic because of the selection of Arabic-speaking nurses:Tool 1: Demographical Information Form: In the present research, this form comprised personal information-based questions for the participants. The collected demographic variables were age, gender, marital status, years of work experience, and workplace.Tool 2: Practice Environment Scale of the Nursing Work Index (Revised, Arabic Version): This scale was developed by Warshawsky and Havens [37] to measure the extent of practice in the work environment on nursing practice, particularly in psychiatric nursing practice in the current study. This study used an already validated Arabic version translated into English [38,39], with permission from the copyright holder dated 01 August 2022. This scale has 32 items and exhibits excellent psychometric properties, with a value as high as 0.87 for relevancy and a reliability of 0.95 for the quality of Arabic translation. It has five subscales: Nurse Participation in Hospital Affairs (NPHA) Subscale; Nursing Foundations for Quality-of-Care (NFQC) Subscale; Nurse Manager Ability, Leadership and Support of Nurses (NMALSN) Subscale; Staffing and Resource Adequacy Subscale (SRAS); and Collegial Nurse–Physician Relationships (CNPR) Subscale. A Likert-type scale that had scores of 1 = strongly agree, 2 = agree, 3 = disagree, and 4 = strongly disagree was used for scoring [38,39].

### 2.5. Statistical Data Analysis

The collected data were analyzed using SPSS version 28 for this research. The following steps were involved in the analysis process:

Data preparation: Data were prepared before statistical analysis could begin. This step involved cleaning the data, checking for missing values, and transforming the data into a suitable format for analysis.

Descriptive statistics: The first step in statistical data analysis was to calculate descriptive statistics, such as measures of central tendency (mean, standard deviation, range, skewness, and kurtosis,) and variability (standard deviation and variance). These statistics provided a summary of the data and helped identify any patterns or trends.

Inferential statistics: In this phase of analysis, this study harnessed a range of statistical tools, including independent *t*-tests, one-way ANOVA, and correlation analysis, to extract meaningful insights about the broader population from this study’s sample data. These methods were thoughtfully selected to cater to the research objectives, which primarily revolved around evaluating mean group differences that pertained to demographic variables. Then, their potential relationships with the study scales were examined. The *t*-test was a valuable choice when the study objective was to discern statistically significant distinctions between the means of two groups, facilitating comparisons, such as gender disparities or variations among different age groups. For scenarios that involved multiple groups, ANOVA proved instrumental in analyzing differences across various categories of a demographic variable. Moreover, correlation analysis was performed to examine the relationship among assessment tool items.

To ensure the integrity of analyses, this study diligently addressed assumptions that underlay these statistical tests. The normality assumption was a crucial consideration for continuous variables, and this study employed methods, such as the Shapiro–Wilk test and visual inspections of normal probability plots, to assess the normal distribution of this study’s data. In instances wherein the data deviated from normality, this study explored suitable transformations or resorted to nonparametric tests, such as the Mann–Whitney U test or the Kruskal–Wallis test, to maintain the robustness of this study’s analyses. Furthermore, this work examined the homogeneity of variance assumption by using tests, such as Levene’s test, which is especially relevant for ANOVA, and sought alternatives, such as Welch’s ANOVA, when homoscedasticity was not met. A meticulous approach to statistical testing and validation aimed to ensure that this study’s inferences were not only statistically sound but also well aligned with the objectives of the research, leveraging the most appropriate methods for the characteristics of this study’s data and the specific research questions.

### 2.6. Ethical Considerations

In the context of this study, the College of Nursing’s Ethical Review Board at King Saud University granted ethical approval (Reference Number: KSU-HE-23-084, dated 17 January 2023). Moreover, the IRBs of Hospital A (Reference Number: H-05-HS-065, dated 14 February 2023) and of Hospital B (Reference Number: NUR002, dated 1 March 2023) gave the approval to collect data from the participants. The ethical review board meticulously finalized its evaluation regarding the study’s measures and methodologies. Data were handled with the utmost confidentiality, with clear assurance that they would solely serve research purposes. The participants were provided with comprehensive explanations, emphasizing their voluntary participation and their right to withdraw from the study at any point without any repercussions. Throughout the research, the following key ethical principles were observed: 

Informed consent: Participants were required to provide informed consent before joining the study. This process entailed the provision of clear and accurate information regarding the study’s purpose, procedures, potential risks, and benefits while obtaining the participants’ voluntary consent. 

Confidentiality and anonymity: Stringent measures were implemented to ensure the confidentiality and anonymity of the participants’ personal information and data, safeguarding them against unauthorized disclosure through the use of pseudonyms or codes to protect the participants’ identities. 

Minimization of harm: Every effort was exerted to minimize potential risks of harm to participants, encompassing physical and psychological aspects. The research sought to ensure that any potential benefits outweighed the risks. 

Respect for participants’ autonomy: The researchers diligently respected participants’ autonomy, assuring them of the right to withdraw from the study at any juncture without the fear of facing adverse consequences. 

Fairness and equity: Commitment was maintained to ensure fairness and equity in the selection of participants, affording everyone an equal opportunity to participate in the study.

## 3. Results

Table 1 provides the frequency and percentages of the demographic profile of the respondents according to which the majority were psychiatric male nurses (*f* = 253, % = 73.1), with an age range of 31–40 years (*f* = 190, % = 54.9), married (*f* = 286, % = 82.7), and working in the Hospital B (*f* = 201, % = 58.1).

In this study, the findings revealed that there were correlations and differences in the practice environment of the participants when grouped according to their socio-demographic characteristics, but the results were not statistically significant. Table 2 indicates the significant correlation between the practice environment scale and the NPHA Subscale (r = 0.923, *p* = 0.01), the NFQC Subscale (r = 0.934, *p* = 0.002), the NMALSN Subscale (r = 0.927, *p* = 0.01), the SRAS (r = 0.855, *p* = 0.001), and the CNPR Subscale (r = 0.868, *p* = 0.003). The NPHA Subscale was significantly correlated with the NFQC Subscale (r = 0.802, *p* = 0.01), the NMALSN Subscale (r = 0.811, *p* = 0.004), the SRAS (r = 0.720, *p* = 0.002), and the CNPR Subscale (r = 0.752, *p* = 0.001). The NFQC Subscale had a significant relationship with the NMALSN Subscale (r = 0.829, *p* = 0.001), the SRAS (r = 0.742, *p* = 0.01), and the CNPR Subscale (r = 0.749, *p* = 0.002). The NMALSN Subscale was significantly associated with the SRAS (r = 0.795, *p* = 0.003) and the CNPR subscale (r = 0.795, *p* = 0.005). The SRAS and the CNPR Subscale (r = 0.767, *p* = 0.001) were significantly correlated. Hence, the findings showed that all the relationships between research variables were positively significant (*p* value less than or equal to 0.01).

In addition, the results indicated that two socio-demographic characteristics were significantly associated with nurses working in the psychiatric department of two mental health hospitals, namely age (r = 0.697, *p* = 0.021) and years of work experience (r = 0.712, *p* = 0.018). The hypothesis was significantly accepted, identifying a significant association between the work environment of nurses working in a psychiatric hospital and their socio-demographic characteristics. The results indicated that the association of the work environment of psychiatric nurses depended on the potential connection between the work environment and affiliation rather than differences in any variable of interest as no significant variations in the perceptions of work environment were revealed when grouped according to participants’ socio-demographic characteristics. The findings also revealed that the factors associated with the work environment had a more significant influence on psychiatric nurses’ overall perception of their work environment than their socio-demographic characteristics.

## 4. Discussion

The present study identified three distinctive research questions based on the aim and its objectives. The first research question involved the effect of the work environment on nurses working in a psychiatric hospital. The results indicated a significant effect of the work environment on nurses working in the psychiatric department. The results align with a cross-sectional study conducted in an Omani public hospital involving 205 nurses indicated that participation in hospital affairs, foundations for quality of care, and staffing and resource adequacy were predictors of burnout and nurses’ perceptions of care quality [40]. Al Sabei and colleagues [40] also revealed that a supportive work environment correlated with reduced intentions to leave the job, depending upon high levels of job satisfaction. Moreover, evidence from previous research that is relevant to the current study indicates that the work environment significantly influences the working conditions and healthcare services provided in hospital settings [41]. Donley [42] reported that a significant and highly effective work environment is required for the appropriate provision of healthcare services, especially in the psychiatric department. Chen and colleagues [43] reported that the working environment is congruent with the working conditions of psychiatric nurses and the rehabilitation services they provide to psychiatric patients. Phillips and colleagues [44] indicated that nurses working in the psychiatric department typically require a smooth and stable environment that seeks appropriate treatment and rehabilitation mechanisms for patient healthcare interventions. Sampaio and colleagues [45] explained that the working condition in the psychiatric department typically depends on the work environment and stability of the nurses working in this department because of the high alert and emergency situation every time a psychiatric hospital faces a patient. The current study uncovers a significant effect of the work environment on nurses working in psychiatric hospitals, which is in line with previous research findings. This study highlights the critical importance of fostering a conducive and stable work environment to facilitate the effective delivery of healthcare services, particularly in psychiatric departments. On the basis of these findings, psychiatric hospitals in the eastern region of the KSA are recommended to prioritize the improvement and maintenance of a supportive work environment for their nursing staff. This step can include strategies for enhancing workplace conditions, promoting teamwork, and providing the resources and support necessary for nurses to deliver high-quality care to psychiatric patients, ultimately contributing to improved patient outcomes and enhanced well-being of healthcare professionals.

Considering the second research question, no association was found between the work environment of psychiatric nurses and the demographic variables. The results presented no significant difference in the demographical variables identifying a potential difference in the working conditions of psychiatric nurses. Foster and colleagues [46] reported that demographical information in identifying gender, age group, marital status, work experience, and hospital were the variables associated with personal information. None of these variables are significantly different from one another when considering the work environment of psychiatric nurses. Varghese [47] determined that demographic information is associated with the identification of personal identity, but it does not significantly affect an individual’s professional life. Furthermore, Al-Maraira and Hayajneh [48] stated that demographic information typically defines the characteristics of an individual. It does not explain the competencies and capabilities of individuals with respect to their working conditions and professionalism. Saleh and colleagues [49] revealed variations in how different demographic groups perceive and experience their work environments. For example, younger nurses may have different expectations and preferences compared with older nurses. Gender-related differences in job roles and responsibilities can also influence the work environment. In addition, years of experience may affect how nurses perceive workplace challenges and opportunities for growth. The findings also indicated no differences or similarities regarding the work environment among the nurses working in the two psychiatric hospitals. The results indicated no significant difference in the work environment of the two mental hospitals in the eastern region of the KSA. The findings showed that the working conditions and the environment of the two psychiatric departments were similar because of psychiatric issues and emergencies. Mukaihata and colleagues [50] stated that the working conditions in a psychiatric department typically depend on the healthcare circumstances and treatment mechanism performed in this department. Wu and colleagues [51] reported that the conditions in psychiatric departments typically influence higher constructivity and stability in the healthcare setting because of the high-alert situation. Sasaki and colleagues [52] argued that the similarities in different departments of psychiatric hospitals depend on the symptoms and consequences of psychiatric illness. Therefore, the similarity of working conditions and environment in psychiatric hospitals usually depends on high reliance on the competencies and capabilities of psychiatric nurses [53,54].

The research findings in this area can inform healthcare organizations and policymakers about the specific needs and concerns of different demographic groups in the healthcare workforce, helping to tailor strategies for improving the work environment and job satisfaction of all employees. Hence, a profound relationship was identified among psychiatric nurses with regard to their work environment, underlining the crucial link between their competencies and skills. Conversely, the hospital settings and demographic factors did not yield significant associations, emphasizing the personalized nature of individual life experiences and constructs. This discussion underscores the paramount importance of comprehending the autonomous influence of working conditions on psychiatric nurses in the eastern region of the KSA, as shaped by the unique challenges posed by psychiatric hospital environments.

Limitations are associated with every study because of research biases to cover a tremendous volume of data and the majority of research paradigms at a spontaneous level. A number of limitations are associated with the present study. The first limitation involves the sample size of 346 nurses working in two different psychiatric hospitals. This sample does not represent the population of psychiatric nurses in the KSA. The second limitation is associated with using the non-probability sampling technique, which involves biases compared with the probability technique. The third limitation is associated with identification using the self-report measurement, which involves social desirability and response bias by the respondent and does not explain the full construct of the work’s effect on the nurses’ working conditions. The fourth limitation is related to the cross-sectional research design. This is associated with the challenges of determining the contribution of different perspectives, which is related to the limitation of establishing the causality variable. The fifth limitation is associated with the limited generalizability and lack of control in quantitative studies; thus, the findings are not generalizable to other hospitals in the KSA.

## 5. Conclusions

The present study indicated a significant association between the work environment of nurses working in psychiatric hospitals and their socio-demographic characteristics. The work environment indicates the condition situation and scenario of a psychiatric hospital with respect to psychiatric nurses’ performance. Moreover, the differences between the two psychiatric hospitals were not significant because of the similarity in psychiatric conditions in psychiatric hospitals. Furthermore, the demographic information regarding psychiatric nurses was not as significant for addressing psychiatric complications associated with psychiatric nurses working in psychiatric hospitals. Overall, this study determined that the work environment in a psychiatric hospital is full of challenges and hardships that require effective and highly appropriate intervention and stability in the environment construct to keep homogenized healthcare services that are easy to maintain by psychiatric nurses.

## 6. Implications and Recommendations

The study’s insights can inform nursing practice by emphasizing the importance of creating supportive and positive work environments. In turn, such environments can contribute to improved patient care and outcomes because nurses working under favorable conditions are likely to provide good quality care. Understanding the factors that contribute to nurse burnout can help healthcare institutions implement strategies for mitigating burnout. By addressing issues such as workplace violence and job dissatisfaction, the well-being of nurses can be safeguarded, leading to more engagement and less burnout among healthcare professionals. Recognizing the value of individualized patient-centered care, as highlighted in this study, can encourage nurses to focus on tailoring their caregiving measures to meet each patient’s unique needs, ultimately improving patient experience. Nursing education programs can incorporate insights from this study to emphasize the significance of workplace engagement, the provision of basic healthcare needs, and individualized patient-centered care. These programs prepare future nurses to be well-rounded professionals who can thrive in diverse healthcare environments. Given the association between workplace violence, job dissatisfaction, and burnout, nursing education can include stress management and coping strategies as essential components of the curriculum. Preparing nurses to handle such challenges can improve their resilience in real-world clinical settings. Policymakers can use the study’s findings to advocate for policies that promote safer and more supportive work environments for nurses. These policies may include regulations to address workplace violence and measures for improving nurse-to-patient ratios. Healthcare institutions can develop and implement support programs to address the specific needs of nurses, particularly in the context of mental health. Policies can be formulated to encourage and facilitate access to these programs. Lastly, regulatory bodies can consider incorporating indicators related to workplace engagement and patient-centered care into quality-of-care standards. Doing so can incentivize healthcare institutions to prioritize these aspects of care in their policies and practices.

Several recommendations are provided to improve the applicability of future research. First, considering a larger sample can enhance the generalizability of the study’s findings. Second, using a stratified sampling strategy can ensure a representative sample of nurses from various hospitals in the area. The sample can be broadened to include nurses from a variety of healthcare settings to better assess the generalizability of the study’s findings. Third, tracking changes in nurses’ perceptions of their work environment over time can be considered by utilizing a longitudinal study that evaluates the impact of interventions aimed at improving the work environment. In addition, to minimize the effect of potential confounders, researchers should consider controlling for factors, such as workload, job satisfaction, and years of experience. This step may influence how their work environment is perceived by nurses.

## Figures and Tables

**Table 1 jcm-13-06506-t001:** Demographical Information of the study participants (N = 346).

Variable	Categories	*f*	%
Gender			
	Male	253	73.1
	Female	93	26.9
Age			
	22–30	74	21.4
	31–40	190	54.9
	41–50	73	21.1
	51–60	9	2.6
Marital status			
	Single	52	15.0
	Married	286	82.7
	Other	8	2.3
Work experience			
	Below 5 years	189	54.6
	5 years and more	157	45.4
Hospital name			
	Hospital A	145	41.9
	Hospital B	201	58.1

Note: *f =* frequency, % = percentage.

**Table 2 jcm-13-06506-t002:** Correlation between research variables (N = 346).

Variables		T_PES	NPHA Subscale	NFQC Subscale	NMALSN Subscale	SRAS	CNPR Subscale
T_PES		1	0.923 **	0.934 **	0.927 **	0.855 **	0.868 **
	NPHA Subscale	-	1	0.802 **	0.811 **	0.720 **	0.752 **
	NFQC Subscale	-	-	1	0.829 **	0.742 **	0.749 **
	NMALSN Subscale	-	-	-	1	0.795 **	0.795 **
	SRAS	-	-	-	-	1	0.767 **
	CNPR Subscale	-	-	-	-	-	1

Note. T_PES = Total of Practice Environmental Scale, NPHA Subscale = Nurse Participation in Hospital Affairs Subscale, NFQC Subscale = Nursing Foundations for Quality-of-Care Subscale, NMALSN Subscale = Nurse Manager Ability, Leadership and Support of Nurses Subscale, SRAS = Staffing and Resource Adequacy Subscale, CNPR Subscale = Collegial Nurse–Physician Relationships Subscale; ** = highly significant at 0.01.

## Data Availability

The data presented in this study are available upon request from the corresponding author.

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
