# Peer review of "Work Environment and Socio-Demographic Factors of Psychiatric Nurses: A Cross-Sectional Study in Hospitals of Eastern Saudi Arabia"

_jcm, 2024, doi:10.3390/jcm13216506_

Round 1

Reviewer 1 Report

Comments and Suggestions for Authors

Dear authors,

Thank you for allowing me to review this article of significant importance which focuses on the The Association between Practice Work Environment and Socio demographic Factors of Psychiatric Nurses in the Eastern Region  Mental Health Hospitals of Saudi Arabia

The title is clear and coherent with the aim and content of the article. I'd just like to suggest the authors, and I fully respect it if, after discussion, the change doesn't make sense to you. I would put the title as follows: Work Environment and Sociodemographic Factors of Psychiatric Nurses: A Cross-Sectional Study in Hospitals of Eastern Saudi Arabia. This change aims to make the title more objective by highlighting the method.

Authors should review the journal's citation and referencing guidelines to revise the article. https://www.mdpi.com/authors/references

 Abstract

I would recommend removing the word background from the beginning of the abstract, as it is underestimated that the initial phase is a framing of the problem. line 11

Lines 11-14, the ideas presented are poorly connected. I recommend revising the text to add sentences and improve the terms used. For example, the sentence in line 11 ‘Dealing with psychiatric patients is not easy.’ is not related to the next idea and does not explain the difficulties. Line 14 ‘of any other individual.’ Is it an individual or are we talking about health professionals? I recommend a general revision of the construction of sentences and terms.

The background only mentions the difficulties of caring for psychiatric patients, but the aim of the work is centered on the work environment. I think it would make sense to highlight the particularities of the work environment of nurses in psychiatric units that make this study relevant, and why it is necessary or innovative.

The information in lines 38-41 could be mobilised in the summary to respond to my comment.

Introduction

I recommend summarising the aspects related to historical details about the Saudi health system that are not directly relevant to the objectives of the study. In addition, the introduction could more clearly highlight the gap in the literature that this study aims to fill.

At the end of the introduction, I don't think point 1.1 is relevant because it only describes the purpose and objectives of the study and no further arguments are presented. I would place this information in the final paragraph of the introduction, because in view of the arguments presented above, it should be clear to the reader why this study, which has the described purpose and the following objectives, should be carried out.

If the intention is to clearly highlight the research questions, one suggestion would be for the authors to mention the research questions directly in the introduction or in another part prior to the discussion, making the structure of the article clearer and more coherent. This would help to better align the sections of the article with the objectives and research questions from the outset, making it easier for readers to understand.

Methods

The authors mention the hospitals involved in the research in detail, specifying the locations (Erada and Mental Health Complex in Dammam and Mental Health Hospital in Al-Ahsa) and providing demographic information on the participating nurses. While this approach is understandable for describing the context of the research, it may jeopardize the confidentiality and anonymity of the participants. As the number of nurses in these hospitals is limited and the demographic data includes variables such as age, gender and years of experience, there is a risk that some participants could be identified, especially by managers or colleagues with access to internal hospital information.

I recommend that the authors consider generalizing the description of the hospitals by referring to them more broadly, such as ‘two psychiatric hospitals in the eastern region of Saudi Arabia’. In addition, I suggest grouping the data more broadly without directly specifying which hospital it is, allowing it to be identified as 1 or 2 or by letters for example. These modifications would help to ensure that the identity of the nurses remains protected while reinforcing the ethical commitment to confidentiality and anonymity of the participants.

Results

The presentation of the results is clear, but the tables could be accompanied by a brief, more interpretative explanation, rather than just listing the statistical values.

Discussion

I recommend including a more detailed comparison with studies carried out in other Middle Eastern countries or in regions with similar mental health structures. This could enrich the discussion by placing the findings within the broader cultural and institutional context.

It would be interesting to include more detailed evidence on which factors of the work environment assessed by the NWI scale varied the most, taking demographic data into account.

The limitations of the study are clearly described.

Conclusion

The conclusion manages to summarise the most important findings, such as the association between the working environment and nurses' performance, and the absence of significant differences between the two hospitals. This provides a good overview of the results.

Implications and Recommendations

The authors provide some practical implications for nursing and suggest recommendations for future research.

References

The references used are current and relevant to the type of study presented.

Author Response

Response to REVIEWER 1 Comments

Dear authors,

Thank you for allowing me to review this article of significant importance which focuses on the The Association between Practice Work Environment and Socio demographic Factors of Psychiatric Nurses in the Eastern Region Mental Health Hospitals of Saudi Arabia

RESPONSE: Dear honorable Reviewer 1, we thank you very much for your valuable comments that helped us in the improvement of our paper. We earnestly hope that we have satisfied all of your comments in our revisions and if not, please know that we are very willing to comply and kindly allow us to do further revisions so that our work will be acceptable for publication in Journal of Clinical Medicine. Our point-by-point responses to each of your comments are indicated below.

The title is clear and coherent with the aim and content of the article. I'd just like to suggest the authors, and I fully respect it if, after discussion, the change doesn't make sense to you. I would put the title as follows: Work Environment and Sociodemographic Factors of Psychiatric Nurses: A Cross-Sectional Study in Hospitals of Eastern Saudi Arabia. This change aims to make the title more objective by highlighting the method.

RESPONSE: Thank you for this helpful comment and we agreed with your suggested title.

Authors should review the journal's citation and referencing guidelines to revise the article. https://www.mdpi.com/authors/references

RESPONSE: Thank you for this comment. In our submitted revised version, we adhered to the journal’s citation and referencing guidelines.

 Abstract

I would recommend removing the word background from the beginning of the abstract, as it is underestimated that the initial phase is a framing of the problem. line 11

Lines 11-14, the ideas presented are poorly connected. I recommend revising the text to add sentences and improve the terms used. For example, the sentence in line 11 ‘Dealing with psychiatric patients is not easy.’ is not related to the next idea and does not explain the difficulties. Line 14 ‘of any other individual.’ Is it an individual or are we talking about health professionals? I recommend a general revision of the construction of sentences and terms.

The background only mentions the difficulties of caring for psychiatric patients, but the aim of the work is centered on the work environment. I think it would make sense to highlight the particularities of the work environment of nurses in psychiatric units that make this study relevant, and why it is necessary or innovative.

The information in lines 38-41 could be mobilised in the summary to respond to my comment.

RESPONSE: Thank you for this comment and for suggesting the information in lines 38-41. We have revised the entire background in Lines 17-24.

Introduction

I recommend summarising the aspects related to historical details about the Saudi health system that are not directly relevant to the objectives of the study.

RESPONSE: Thank you for this comment. We revised these aspects in Lines 66-83.

In addition, the introduction could more clearly highlight the gap in the literature that this study aims to fill.

RESPONSE: Thank you for this comment. We covered this in Lines 156-172.

At the end of the introduction, I don't think point 1.1 is relevant because it only describes the purpose and objectives of the study and no further arguments are presented. I would place this information in the final paragraph of the introduction, because in view of the arguments presented above, it should be clear to the reader why this study, which has the described purpose and the following objectives, should be carried out.

RESPONSE: Thank you for this comment. We covered this in Lines 173-187.

If the intention is to clearly highlight the research questions, one suggestion would be for the authors to mention the research questions directly in the introduction or in another part prior to the discussion, making the structure of the article clearer and more coherent. This would help to better align the sections of the article with the objectives and research questions from the outset, making it easier for readers to understand.

RESPONSE: Thank you for this comment. We followed your suggestion by mentioning this in the last paragraph of the Introduction section. In addition, we added two hypotheses in Lines 188-192, as suggested by honorable Reviewer 2.

Methods

The authors mention the hospitals involved in the research in detail, specifying the locations (Erada and Mental Health Complex in Dammam and Mental Health Hospital in Al-Ahsa) and providing demographic information on the participating nurses. While this approach is understandable for describing the context of the research, it may jeopardize the confidentiality and anonymity of the participants. As the number of nurses in these hospitals is limited and the demographic data includes variables such as age, gender and years of experience, there is a risk that some participants could be identified, especially by managers or colleagues with access to internal hospital information.

I recommend that the authors consider generalizing the description of the hospitals by referring to them more broadly, such as ‘two psychiatric hospitals in the eastern region of Saudi Arabia’. In addition, I suggest grouping the data more broadly without directly specifying which hospital it is, allowing it to be identified as 1 or 2 or by letters for example. These modifications would help to ensure that the identity of the nurses remains protected while reinforcing the ethical commitment to confidentiality and anonymity of the participants.

RESPONSE: Thank you for this comment. We followed one of your suggestions and indicated that Al-Ahsa Mental Health Hospital as Hospital A and Erada and Mental Health Complex in Dammam as Hospital B throught the main text. But, we did not follow this revision in the  Institutional Review Board Statement in Lines 583-588.

Results

The presentation of the results is clear, but the tables could be accompanied by a brief, more interpretative explanation, rather than just listing the statistical values.

RESPONSE: Thank you for this comment. We covered this in Lines 392-407.

Discussion

I recommend including a more detailed comparison with studies carried out in other Middle Eastern countries or in regions with similar mental health structures. This could enrich the discussion by placing the findings within the broader cultural and institutional context.

RESPONSE: Thank you for this comment. We added this in Lines 435-440.

It would be interesting to include more detailed evidence on which factors of the work environment assessed by the NWI scale varied the most, taking demographic data into account.

RESPONSE: Thank you for this comment. We covered this in Lines 416-428. For this part, we are currently unable to provide an additional table and if required, we will comply.

The limitations of the study are clearly described.

RESPONSE: Thank you so much for this comment.

Conclusion

The conclusion manages to summarise the most important findings, such as the association between the working environment and nurses' performance, and the absence of significant differences between the two hospitals. This provides a good overview of the results.

RESPONSE: Thank you so much for this comment.

Implications and Recommendations

The authors provide some practical implications for nursing and suggest recommendations for future research.

RESPONSE: Thank you so much for this comment.

References

The references used are current and relevant to the type of study presented.

RESPONSE: Thank you so much for this comment.

Reviewer 2 Report

Comments and Suggestions for Authors

The article presents solid research that addresses a relevant and timely topic. The focus on the influence of the work environment on the job satisfaction of psychiatric nurses is pertinent and provides important data to improve both the working conditions of these professionals and the quality of patient care. The methodological design, based on a cross-sectional and quantitative study, is well justified, and the results obtained reinforce the importance of providing a supportive work environment for healthcare professionals. Additionally, the article is backed by an exhaustive literature review, which places the study within an appropriate academic context.

Some improvements that should be made are as follows: Although the article clearly outlines the objectives, a more specific section detailing the hypotheses would enhance the understanding of the central focus of the research. For example, an explicit hypothesis could be formulated regarding the differential influence of demographic characteristics on job satisfaction.

Furthermore, the discussion would benefit from a deeper analysis that not only compares the findings with the existing literature but also explores the implications of these results in greater detail. More specific practical recommendations could be included for hospital managers and healthcare policymakers.

Consider incorporating additional graphs and tables to more clearly illustrate the significant relationships between the variables found. This would help facilitate the interpretation of the results and provide added value in terms of visual presentation.

Since the existence of differences in nurses' perceptions of the work environment according to their age and experience is mentioned, it might be interesting to conduct a subgroup analysis to evaluate how these variables influence different aspects of the work environment (e.g., leadership, resources, support).

The article acknowledges some limitations, but a more detailed discussion on how these limitations may have influenced the results and how future studies could overcome them would be useful. Additionally, you could consider including more specific suggestions for future research, such as a longitudinal study that evaluates the impact of interventions aimed at improving the work environment.

Author Response

Response to REVIEWER 2 Comments

The article presents solid research that addresses a relevant and timely topic. The focus on the influence of the work environment on the job satisfaction of psychiatric nurses is pertinent and provides important data to improve both the working conditions of these professionals and the quality of patient care. The methodological design, based on a cross-sectional and quantitative study, is well justified, and the results obtained reinforce the importance of providing a supportive work environment for healthcare professionals. Additionally, the article is backed by an exhaustive literature review, which places the study within an appropriate academic context.

RESPONSE: Dear honorable Reviewer 2, we thank you very much for your valuable comments that helped us in the improvement of our paper. We earnestly hope that we have satisfied all of your comments in our revisions and if not, please know that we are very willing to comply and kindly allow us to do further revisions so that our work will be acceptable for publication in the Journal of Clinical Medicine. Our point-by-point responses to each of your comments are indicated below.

Some improvements that should be made are as follows: Although the article clearly outlines the objectives, a more specific section detailing the hypotheses would enhance the understanding of the central focus of the research. For example, an explicit hypothesis could be formulated regarding the differential influence of demographic characteristics on job satisfaction.

RESPONSE: Thank you for this comment. We added this in Lines 188-192.

Furthermore, the discussion would benefit from a deeper analysis that not only compares the findings with the existing literature but also explores the implications of these results in greater detail.

RESPONSE: Thank you for this comment. We covered this at the end of paragraph one of the Discussion section (Refer to Lines 454-464). Also, in paragraph two (Refer to Lines 478-482) and in paragraph three (Refer to Lines 498-508) of the Discussion section.

More specific practical recommendations could be included for hospital managers and healthcare policymakers.

RESPONSE: Thank you for this comment. We covered this in the Implications and Recommendations section (Refer to Lines 555-563).

Consider incorporating additional graphs and tables to more clearly illustrate the significant relationships between the variables found. This would help facilitate the interpretation of the results and provide added value in terms of visual presentation.

RESPONSE: Thank you for this comment. We covered this in Lines 416-428, but we sincerely apologize that we are currently unable to provide an additional table due to time constraints. However, if required and advised by you and the academic editor, we shall willingly comply with a new resubmission due date.

Since the existence of differences in nurses' perceptions of the work environment according to their age and experience is mentioned, it might be interesting to conduct a subgroup analysis to evaluate how these variables influence different aspects of the work environment (e.g., leadership, resources, support).

RESPONSE: Thank you for this comment. Also, for this, we sincerely apologize that we are currently unable to provide an additional analysis due to time constraints. However, if required and advised by you and the academic editor, we shall willingly comply with a new resubmission due date.

The article acknowledges some limitations, but a more detailed discussion on how these limitations may have influenced the results and how future studies could overcome them would be useful. Additionally, you could consider including more specific suggestions for future research, such as a longitudinal study that evaluates the impact of interventions aimed at improving the work environment.

RESPONSE: Thank you for this suggestion. We covered this in Lines 570-571.

Round 2

Reviewer 1 Report

Comments and Suggestions for Authors

Dear authors,

I would like to thank you for your efforts in taking a detailed account of the comments and suggestions made in the review. The changes made are comprehensive and demonstrate a commitment to the quality and clarity of the article.

The new title has become more objective, aligning well with the content and method used. The revised abstract and introduction adequately addressed the points raised, clearly highlighting the relevance and justification of the study. The concern for the confidentiality of the participants in the Methods section was well received, and the modifications made to the hospitals guarantee greater anonymity without compromising the integrity of the study.

Overall, the changes made were satisfactory and meet the recommendations. This version is ready to move forward in the publication process.

Thank you very much for the opportunity to review your work.